# Dietary Supplementation with Omega-3 Polyunsaturated Fatty Acids Reduces Opioid-Seeking Behaviors and Alters the Gut Microbiome

**DOI:** 10.3390/nu11081900

**Published:** 2019-08-14

**Authors:** Joshua K. Hakimian, Tien S. Dong, Jorge A. Barahona, Venu Lagishetty, Suchi Tiwari, Darien Azani, Matthew Barrera, Suhjin Lee, Amie L. Severino, Nitish Mittal, Catherine M. Cahill, Jonathan P. Jacobs, Wendy M. Walwyn

**Affiliations:** 1Department of Psychiatry and Biobehavioral Sciences, Hatos Center for the Study of Opioids Receptors and Drugs of Abuse, UCLA Semel Institute for Neuroscience and Human Behavior, David Geffen School of Medicine, UCLA, Los Angeles, CA 90095, USA; 2The Vatche and Tamar Manoukian Division of Digestive Diseases, Department of Medicine, David Geffen School of Medicine at UCLA, Los Angeles, CA 90095, USA; 3UCLA Microbiome Center, David Geffen School of Medicine at UCLA, Los Angeles, CA 90095, USA; 4Division of Pharmacology and Toxicology, College of Pharmacy, University of Texas at Austin, Austin, TX 78712, USA; 5ZS Associates, San Mateo, CA 94402, USA; 6Division of Gastroenterology, Hepatology and Parenteral Nutrition, VA Greater Los Angeles Healthcare System, Los Angeles, CA 90025, USA; 7UCLA Brain Research Institute, UCLA, Los Angeles, CA 90095, USA

**Keywords:** opioid, microbiome-brain axis, DHA, EPA, anxiety, polyunsaturated fatty acids, Intravenous self-administration, mice

## Abstract

Opioids are highly addictive substances with a relapse rate of over 90%. While preclinical models of chronic opioid exposure exist for studying opioid dependence, none recapitulate the relapses observed in human opioid addiction. The mechanisms associated with opioid dependence, the accompanying withdrawal symptoms, and the relapses that are often observed months or years after opioid dependence are poorly understood. Therefore, we developed a novel model of chronic opioid exposure whereby the level of administration is self-directed with periods of behavior acquisition, maintenance, and then extinction alternating with reinstatement. This profile arguably mirrors that seen in humans, with initial opioid use followed by alternating periods of abstinence and relapse. Recent evidence suggests that dietary interventions that reduce inflammation, including omega-3 polyunsaturated fatty acids (n-3 PUFAs), may reduce substance misuse liability. Using the self-directed intake model, we characterize the observed profile of opioid use and demonstrate that an n-3-PUFA-enriched diet ameliorates oxycodone-seeking behaviors in the absence of drug availability and reduces anxiety. Guided by the major role gut microbiota have on brain function, neuropathology, and anxiety, we profile the microbiome composition and the effects of chronic opioid exposure and n-3 PUFA supplementation. We demonstrate that the withdrawal of opioids led to a significant depletion in specific microbiota genera, whereas n-3 PUFA supplementation increased microbial richness, phylogenetic diversity, and evenness. Lastly, we examined the activation state of microglia in the striatum and found that n-3 PUFA supplementation reduced the basal activation state of microglia. These preclinical data suggest that a diet enriched in n-3 PUFAs could be used as a treatment to alleviate anxiety induced opioid-seeking behavior and relapse in human opioid addiction.

## 1. Introduction

Several factors converged in the early 2000s to contribute to the escalating opioid epidemic. These included an over-prescription of potent and synthetic opioids, a belief that chronic pain was protective against the development of addictive behavior, an aggressive marketing strategy by the manufacturers, and the incorrect translation that long-term use of extended release opioids, safe in terminally-ill cancer patients, could be used in non-cancer patients without caution [1]. Alarming statistics from recent years document the increase in mortality from the use of fentanyl and other synthetic opioids that are often prescribed for pain [2,3,4,5]. For many of these cases, the initial exposure to opioids began with oxycodone and other prescription analgesics, and was then transferred to other more rapidly acting opioids [6]. We demonstrate in this paper that rapidly acting opioids are powerfully reinforcing and that dietary supplementation of omega 3 polyunsaturated fatty acids (n-3 PUFAs) may reduce the ability of these rapid-acting and potent compounds to maintain addictive-like behaviors via action on gut microbiome composition. 

Whichever the opioid, an inherent property of these addictive substances is the high rate (≈91%) of relapse [7] and the relative lack of effective treatment [8]. The rate of relapse following chronic opioid use has been linked to allostatic mechanisms that maintain persistent drug seeking such as a decline in cognitive control over habituated behaviors and tolerance to drug effects over time. The mechanisms that produce these long-lasting addictions are complex and can induce cravings that lead to relapse months or years after the physical opioid dependence is no longer a factor. According to the Impaired Response Inhibition and Salience Attribution (iRISA) model of addiction proposed by Rita Goldstein and colleagues, the increased salience of drug-paired cues and impaired response inhibition in the absence of the drug contribute to increased drug seeking over time and a persistence of seeking in the absence of the drug [9,10,11]. These physical and emotional changes result in a persistent negative emotional state culminating in an inability to transition to an opioid-free state and to remain opioid-free [12]. Coupled with an increased vulnerability to stress and both physical and psychological stressors that trigger drug use in drug addicted humans and animals [13], relapse after prior opioid use is a common occurrence that must be addressed. 

Several studies have focused on the bidirectional communication that takes place between the gastrointestinal tract and the central nervous system (CNS). Previous studies have revealed the major role gut microbiota have on brain function, and consequently, neuropathology [14,15,16,17,18]. Anxiogenic affective-like behaviors are lower in mice that exhibit increases in specific genera, including *Lactobacillus* and *Bifidobacterium*, and these changes were linked to altered neural function [19,20]. A similar relationship between *Bifidobacterium* and anxiety in humans has been established [21]. Still, the role of nutrition and supplementation in the microbiome-brain axis in psychiatry is not fully understood. 

The family of n-3 PUFAs consists of docosahexaenoic acid (DHA; c22:6n-3), ecosapentaenoic acid (EPA; c20:5n-3), both available from fish, and alpha-linolenic acid (ALA: c18; 3n-3), available from plants. The beneficial effects of supplementary n-3 PUFAs for many conditions and diseases have been the subject of ongoing research. A recurrent observation is that dietary omega-3 supplementation relieves anxiety and depression symptoms in mood disorders often co-diagnosed in substance abuse disorder patients (for reviews, see References [22,23,24]). Accordingly, n-3 PUFA supplements enriched in DHA have been shown to reduce anxiogenic affective-like behaviors in preclinical models [22,25,26] and clinical trials [23,27,28,29,30,31]. DHA and EPA have both shared independent anti-inflammatory properties by intervening at different points within the inflammatory cascade, which provides an important consideration for their translational benefits (reviewed in Calder [32]). DHA is also required as a structural component of plasma, microsomal, and synaptic membranes in the brain [33,34], and is essential for numerous brain functions including development [35] and diverse cellular functions [36,37,38,39,40]. There is also evidence that n-3 PUFA supplementation influences the gut microbiota, which can contribute to changes in brain function [41,42,43]. The overarching hypothesis for our study was that n-3 PUFA intake induces a beneficial shift in gut microbiome composition to normalize the genera that become overgrown with opioid-self administration. 

We have previously shown that chronic, noncontingent morphine induces diverse adaptations of the glutamatergic system in the striatum, a hub of the reward-centered mesolimbic circuitry [44]. Many of these cellular and behavioral adaptations were reduced or offset by a PUFA-enriched diet. The model of opioid exposure used in this previous study was an escalating dose of morphine, administered over 5 days via subcutaneous injection twice daily, followed an additional 3 days of morphine (at the highest dose), a noncontingent method of delivery. In the present study, we used an opioid exposure model with greater predictive and face validity to mimic opioid use disorder. This model uses contingent exposure whereby there is a choice as to how much opioid to self-administer through an indwelling intravenous catheter. The delivery of each drug infusion was contingent on pressing a designated active lever in an operant box. This allowed us to generate a 27-day profile of opioid use in mice, equating to 3 years of a human life [45], beginning with the acquisition of this behavior, followed by a period of maintenance, then periods of extinction alternating with reinstatement. This profile of opioid self-administration arguably recapitulates that seen in humans, with initial opioid use followed by alternating periods of abstinence and relapse. Using this model, we first outline this opioid use profile and then asked to what extent an n-3-PUFA-enriched diet may alter drug-seeking behaviors. We also assessed to what extent this supplementation had on affective-like behavior using the light–dark test. We then examined how chronic opioid self-administration and the n-3 PUFA supplementation, separately and in combination, altered gut microbiome composition. We achieved this by examining the gut microbiome during multiple phases of the opioid exposure model. Lastly, we examined the state of the microglia in the striatum, a site we had previously studied [43], at the completion of the trial to determine whether there was evidence of altered microglial activation due to the opioid exposure or n-3 PUFA supplementation. 

## 2. Materials and Methods

All the experiments were conducted in accordance with the American Association for Laboratory Animal Care (AALAC) Guide for the Care and Use of Laboratory Animals and approved by the University of California Los Angeles (UCLA) Institutional Animal Care and Use Committee (IACUC committee) (Office of Animal Research Oversight (OARO) #1999-179). Mice (C57Bl6/J Jax stock # 000664), 6–8 weeks of age at the start of the experiment, were randomly assigned to a control lab chow diet containing 0.5% DHA (control group) alone or supplemented with 2.5% DHA, 1.1% EPA, and 0.75% other omega-3 PUFAs (Nordic Naturals, Watsonville, CA, USA) for 8 weeks (n-3 PUFA group). We have previously shown an increase in DHA enrichment in specific brain regions in male, but not female, C57Bl6/J mice following this same protocol [44]. Accordingly, only male C57Bl6/J mice were used for these experiments. Of note is that as the PUFA supplementation protocol carried a strong fish odor; as such, it was not possible for experimenters to remain blind to this intervention for the behavioral procedures. However, experimenters were blind for all other procedures and data analyses. 

### 2.1. Intravenous Self-Administration (IVSA) Administration

Mice were divided into four groups: control diet + saline (*n* = 14), control diet + opioid (*n* = 14), n-3 PUFA diet + saline (*n* = 5), and n-3 PUFA diet + opioid (*n* = 10). An intravenous catheter (0.2 mm i.d., 0.4 mm o.d., NorfolkAccess, Skokie, IL, USA) was inserted into the right jugular vein of mice under sterile conditions, as we have previously described [46,47,48]. Post-operative care included Carprofen gel food (MediGel CPF, Portland, ME, USA). Catheters were flushed daily with 0.02 mL Heparin/saline (30 USP/mL). The overarching timeline of experiments is described in Figure 1. Mice were monitored twice daily for the first 48 h after surgery and exclusion criteria of more than a 15% weight lost or a moribund state for over 24 h were applied. Catheter patency was tested using an infusion of propofol (20 μL of 1% propofol *w*/*v* in saline) every five days. After 3 days of postoperative recovery, mice were trained to lever press in the self-administration operant boxes (Med-Associates Georgia, VT, USA) by using a droplet of 20% sweetened condensed milk placed above both the active (AL) and inactive (IAL) levers (3times per session) during the first two 120 min sessions. Mice underwent daily 2-h self-administration sessions where the active lever was paired with remifentanil and an inactive lever was paired with a saline infusion using a random lever-infusion assignment (Figure 1). An AL press resulted in an intravenous drug infusion (0.67 μL/g body weight) and the presentation of a 10-s tone and visual cue (light); together, these comprised a “reinforcer” (RNFS). A 10-s “timeout” period followed each RNFS, during which time, no further RNFSs could be earned. The mice first underwent a minimum of 3 days of acquisition training using remifentanil (0.05 mg/kg/infusion) as the delivered opioid at a fixed ratio of one drug infusion for each lever press (FR1) to a maximum of 50 RNFS or 2 h, whichever came first (the acquisition phase). Once the criterion of 50 RNFS or 20% stability was obtained for two consecutive days, the mice were transitioned to oxycodone (0.25 mg/kg/infusion) under the same short-access 2-h FR1 schedule for 10 days (the maintenance phase). This was followed by 5 days of extinction (1E), in which no further drug was delivered in response to an AL press but the same environment and reward-associated cue conditions were presented; however, there was no limit on the number of RNFSs that could be earned, with the session ending after 2 h. This was followed by a 2-day period of reinstatement in which oxycodone was administered intravenously at the same dose and cues as used during maintenance. Thereafter the mice underwent a second extinction for an additional 5 days under the same conditions as during the first extinction with no limit on the number of RNFSs that could be earned and the session ended after 2 h (2E). Due to the loss of catheter patency, the number of mice in these later stages declined from those at the beginning of the experiment.

Statistical Analyses. The inter- and intra-session data were analyzed as a function of group and time using a linear mixed model in R (R Core Team, 2015) using the package “lmerTest” [49]. The linear models were generated for each experimental group of interest. The models were used to assess the effect of time, diet group, or an interaction of these factors on AL presses and RNFSs earned. Whenever a significant effect was observed, a new reduced model was generated by removing the significant factor and compared with the original model using an ANOVA in order to assess the impact of the respective factor on the goodness-of-fit for the model. The resulting model was a regression equation where the intercept was allowed to vary for each subject: Y_Characteristic_ = β_0_ + β_Group_X_Group_ × β_Time_X_Time_ + U_Subject_, where Y_Characteristic_ is the characteristic being modeled (e.g., active lever presses, reinforcers earned), each predictor variable is represented by its subscripted X, and U_Subject_ represents the random effect of each individual subject. The coefficients (β) were estimated and assessed for significance, and the contribution to the goodness of fit of the model was assessed.

### 2.2. Light–Dark Assay

The apparatus used for this test of anxiety-related behavior [50] consisted of light and dark compartments of a square box (28 cm^2^ square 18 cm height) separated by a guillotine door in a quiet room illuminated at 50–55 lux. The light compartment was illuminated at 1000 lux by a halogen lamp, measured at the guillotine door. Mice (control diet, *n* = 7; n-3 PUFA diet, *n* = 7) were placed in the dark compartment with the door closed for 2 min, after which, the guillotine door was raised to allow free movement between the light and dark compartments for an additional 2 min. Video tracking was recorded using a A1300-60gm Basler ace camera (106580-08, Basler, Germany), saved by the downloadable VLC video software (VideoLAN), and analyzed for the time for a full body exit from the dark compartment by experimenters blind to diet and opioid treatments.

Statistical Analysis. Data were analyzed using linear mixed models as previously described [49]. This test was conducted 3 days before insertion of the intravenous catheter and just before the operant session on the second day of the second extinction period (25 days after the initial test).

### 2.3. Microbiome Characterization: 16S Ribosomal RNA Sequencing

Stool collection was done at the following time points: baseline (control diet, *n* = 5; n-3 PUFA diet, *n* = 3), first day of oxycodone IVSA (control diet, *n* = 11; n-3 PUFA diet, *n* = 11), 10th day oxycodone IVSA (control diet, *n* = 15; n-3 PUFA diet, *n* = 10), fifth day of first extinction (control diet, *n* = 15; n-3 PUFA diet, *n* = 10), and the first day of the second extinction (control diet, *n* = 7; n-3 PUFA diet, *n* = 7). Stool was collected a minimum of 30 min before any behavioral tests, including IVSA. Stool was collected fresh and did not interact with any non-sterile surfaces. Mice were placed on top of a cage (to provide them with something to hold onto), held gently at the base of their tails and massaged until stool started to be naturally expelled. Once stool was partly exposed, sterile forceps were used to remove the stool and place it into a sterile tube. Within 2 min of extraction, they were placed in an −80 °C freezer and stored for 2–3 months prior to 16S rRNA sequencing. DNA was extracted from frozen fecal pellets using the PowerSoil DNA Isolation Kit (MO BIO Laboratories, Carlsbad, CA, USA) with bead beating following the manufacturer’s instructions. The V4 region of 16S ribosomal RNA genes was amplified and underwent 2 × 150 sequencing on an Illumina MiSeq (Illumina, San Diego, CA, USA) as previously described [51]. The base pair reads were processed using QIIME v1.9.1 with default parameters [52]. The sequence depth ranged from 47,843 to 128,236 sequences per sample. Operational taxonomic units (OTUs) were picked against the May 2013 version of the Greengenes database, pre-filtered at 97% identity. OTUs were removed if they were present in less than 10% of samples. Alpha diversity (i.e., diversity within a sample) and beta diversity (differences in composition across samples) were calculated in QIIME using OTU-level data rarefied to 47,843 sequences.

Statistical Analyses. The significance of differences in alpha diversity metrics—Faith’s phylogenetic diversity (Faith’s PD), Chao1, and Shannon index—was calculated using analysis of variance. Beta diversity was calculated using Bray-Curtis dissimilarity and visualized using principal coordinates analysis. Adonis, a permutational analysis of variance, was performed using 100,000 permutations to test for differences in distances across diet and groups [53,54]. Association of microbial genera with diet and group (oxycodone and extinction phase) were evaluated using DESeq2 in R, which employs an empirical Bayesian approach to shrink dispersion and fit non-rarified count data to a negative binomial model [54]. It was seen that a few mice in the colony were affected by an overgrowth of segmented filamentous bacteria (i.e., *Candidatus Arthromitus*). This species only contributed to less than 0.1% of the overall relative abundance, and because this represented a contaminant genus unrelated to the experimental interventions, it was excluded from beta-diversity, alpha-diversity, and DESeq2 analysis. The *p*-values for differential abundance were converted to *q*-values to correct for multiple hypothesis testing (<0.05 for significance) [55].

### 2.4. IBA1 and CD68 Labeling

Brains from the four testing groups, including the control diet + saline (*n* = 2), control diet + oxycodone (*n* = 2), n-3 PUFA diet + saline (*n* = 2), and n-3 PUFA diet + oxycodone (*n* = 5), were collected 48 h after the last operant session. They were then placed in 4% paraformaldehyde overnight at 4 °C, followed by 24–48 h in 30% sucrose until equilibrated, then frozen in a dry ice/isopropanol bath and stored at −80 °C. Fifty micrometer sections were later cut and collected into phosphate buffered saline (PBS), washed in PBS with 0.1% Triton X-100 (PBS-T; Sigma, city, MO, USA) 3 × 10 min each, and blocked for 2 h at room temperature (RT) in 5% NGS + 3% BSA in PBS-T. They were then incubated in the following antibodies diluted in the wash buffer; anti-rabbit IBA-1 (1:2000; Wako, Richmond, VA, USA; cat. # 019-19741) and anti-mouse CD68 (1:1000; BioRad Hercule, CA, USA; cat. # MCA 1957) overnight at 4 °C. After another three washes for 10 min each, sections were incubated in the following secondary antibodies; goat anti-rabbit 488 (1:1000) and goat anti-Rat 647 (1:1000; both from Thermofisher, Waltham, MA, USA), diluted in PBS-T for 2 h at RT. After a final two washes in PBS-T and one in PBS for 10 min at RT, the sections were mounted on Permamount slides and coverslipped with a DAPI Antifade Mounting Medium (VectorLabs, Burlingame, CA, USA). A Leica DM5500 B upright microscope with a Leica DFC9000 GT sCMOS camera and LAS X software (Lieica, Germany) and a 20× objective was used to obtain tiled images of each section. Sections (10 × 10”) from dorsomedial and dorsolateral regions of the striatum were exported to ImageJ [56].

*Statistical Analyses.* The shape and intensity of CD68 labeled cells was quantified by a user blind to the group identity with 5–30 cells quantified in each image and 5–10 images quantified for each of the dorsolateral and dorsomedial striatal sections of each mouse, and as region had no effect, they were combined into a single dataset for analysis using one-way ANOVA (Prizm v8). 

## 3. Results

### 3.1. The Experimental Timeline

A schematic of the timeline used is shown in Figure 1 and described as follows. The basal light–dark test and fecal collection occurred prior to the IVSA surgery, which was then followed by the IVSA self-administration protocol consisting of five sequential phases; acquisition, maintenance, extinction, reinstatement, and a second extinction. Fecal collection was performed between each of these phases with the final light–dark test taking place on day 24 during the second extinction. The experiment terminated with the collection of brain tissue on day 27 (Figure 1). 

### 3.2. An n-3 PUFA-Enriched Diet Reduced Oxycodone-Seeking Behaviors

To both characterize the opioid self-administration profile and to assess the effect of an n-3 PUFA-enriched diet on this profile, mice maintained on a standard laboratory chow diet were compared to mice given an increased level of n-3 PUFAs supplemented into the food. Both groups of mice were trained to self-administer opioids in order to analyze drug-specific behaviors, the first when remifentanil was self-administered (Appendix A) and the second when oxycodone was self-administered (Figure 2A–F).

Mice underwent at least three days of acquisition training using remifentanil as the delivered opioid on an FR1 schedule. There was no effect of diet in the initial acquisition of opioid self-administration; using remifentanil as the reinforcer, neither active lever presses (F_(2,88)_ = 1.185, *p* = 0.3) nor reinforcers (F_(2,88)_ = 0.837, *p* = 0.44) were significantly different between the diet groups (Appendix A).

The number of AL presses across this profile showed an effect of day (*p* < 0.001, χ^2^ = 12.147) and diet (*p* < 0.05, χ^2^ = 4.646), as well as a significant day-by-diet interaction (*p* < 0.01, χ^2^ = 7.526). For mice on the control diet, post-hoc analysis revealed an increase in AL pressing on 1E1 (*p* < 0.0001, χ^2^ = 21.367) and a decline in lever pressing thereafter during sessions 2E2–5 (*p* < 0.0001, χ^2^ = 31.481). AL pressing during reinstatement did not differ from the last day of the first extinction period (1E), but again increased on the first day of 2E following the 2-day reinstatement of oxycodone (R1 or R2 vs. 2E1; *p* < 0.01, χ^2^ = 7.164). For mice on the n-3 PUFA diet, post-hoc analysis showed an initial similarity to those on the control diet with an increase in lever pressing on 1E1 (*p* < 0.01, χ^2^ = 10.583) and a decline in lever pressing for each day of 1E thereafter (*p* < 0.0001, χ^2^ = 20.022). However, the second extinction period following the two reinstatement sessions did not induce an increase in ALs. Furthermore, AL pressing across sessions 2E1–5 by the n-3 PUFA mice was lower than that of the control mice (*p* < 0.05, χ^2^ = 5.136) (Figure 2A). These results are consistent with the n-3 PUFA-enriched diet reducing oxycodone seeking behaviors during the second but not first extinction period.

In assessing goal-directed lever pressing behavior, the number of inactive (no drug infusion) lever presses showed significant day-by-diet interaction (*p* < 0.05, χ^2^ = 4.775); however, there was no independent effect of diet, (*p* = 0.70, χ^2^ = 0.143) or day (*p* = 0.58, χ^2^ = 0.305) on this parameter (Figure 2B). Goal-directed lever pressing behavior can also be evaluated by the percent of active lever presses out of the total lever presses (AL/(AL + IAL)). This showed an effect of day (*p* < 0.0001, χ^2^ = 19.231), but not diet (*p* = 0.67, χ^2^ = 0.178), nor any day-by-diet interaction (*p* = 0.69, χ^2^ = 0.156) (Figure 2C).

A press on the AL followed by the delivery of oxycodone or saline, a tone and a 10-s visual light cue, and a 10-s time-out period together comprised the reinforcer (RNFS). When measuring the number of RNFSs earned, we found that this parameter mirrored the AL presses in that there was a day (*p* < 0.01, χ^2^ = 10.361) and diet effect (*p* < 0.05, χ^2^ = 4.694), as well as a day-by-diet interaction (*p* < 0.01, χ^2^ = 9.726). Further post-hoc analysis showed that, for mice on the control diet, the number of RNFSs earned increased on 1E1 (*p* < 0.0001, χ^2^ = 16.547) and declined thereafter (*p* < 0.0001, χ^2^ = 25.288). The number of RNFSs obtained during reinstatement did not differ from the last day of E1 and were lower than the last day of maintenance (*p* < 0.01, χ^2^ = 10.538). However, there was an increase in RNFSs earned on the first day of the second extinction period (2E1) above that of reinstatement (R1 vs. 2E1; *p* < 0.01, χ^2^ = 7.790). For mice on the n-3 PUFA diet, post-hoc analysis revealed an initial similarity with those on the control diet, showing an increase in RNFSs earned on 1E1 (*p* < 0.0001, χ^2^ = 17.498) and a decline in RNFSs for each day of 1E thereafter (*p* < 0.0001, χ^2^ = 20.478). However, the 2E1–5 sessions did not induce an increase in RNFSs earned when compared to the level observed during reinstatement. Furthermore, the RNFSs earned by the n-3 PUFA mice across the five days of this 2E period were lower than that of the control mice (*p* < 0.01, χ^2^ = 6.842) (Figure 2D).

Given our observation that the first day of 2E showed a clear protective effect for the n-3 PUFA diet on opioid seeking behaviors, we then focused on the within-session behavior. Specifically, we analyzed the rate of AL pressing or RNFSs earned within the 2-h access window on the first day of 2E. Using a mixed models linear analysis of these datasets, we found that the rate of AL pressing showed an effect of diet (*p* < 0.05, χ^2^ = 3.889), time (*p* < 0.0001, χ^2^ = 30.788), and an interaction of diet and time (*p* < 0.01, χ^2^ = 6.800) with the control group pressing more frequently than the n-3 PUFA group (Figure 2E,F). Further in-depth analysis of the hourly data showed an effect of diet (*p* < 0.05, χ^2^ = 4.018), but not time, on the frequency of AL pressing with the control group pressing more during the first hour. During the second hour, the frequency of AL pressing declined, resulting in an effect of time (*p* < 0.05, χ^2^ = 4.155), and a diet-by-time interaction (*p* < 0.05, χ^2^ = 4.884), but no effect of diet alone (Figure 2E). In contrast, the rate of RNFSs earned did not show any clear effect of diet, time, nor an interaction between the control and n-3 PUFA diet for the entire 2-h session. Post-hoc analysis of each hour revealed no significant difference in RNFSs earned during the first hour. However, there was an effect of diet (*p* < 0.01, χ^2^ = 7.254), time (*p* < 0.01, χ^2^ = 8.041), and a diet-by-time interaction (*p* < 0.001, χ^2^ = 10.993) for the second hour. This was due to the rate of RNFSs earned by the n-3 PUFA mice declining more rapidly than the control group during this hour (Figure 2F).

### 3.3. The n-3 PUFA-Enriched Diet Reduced Anxiety-Like Behavior during Extinction

We have previously shown that an n-3 PUFA-enriched diet reduces the anxiety state of mice following chronic morphine [44]. Here we used the well-established light–dark test to repeatedly assess the state of anxiety [50,57] (Figure 2G). Using linear regression analysis, we found that although the n-3 PUFA diet did not alter the basal levels of anxiety (*p* = 0.4845, F_(1,12)_ = −0.5203), the n-3 PUFA-enriched diet reduced anxiety-like behavior as assessed by decreased latency into the anxiogenic light compartment (*p* < 0.05, F_(1,12)_ = 7.027).

### 3.4. Microbiome Profiles Changed with an n-3 PUFA-Enriched Diet and with Opioid Extinction

We characterized the effect of n-3 PUFA on gut microbiome composition in the context of opioid exposure by performing 16S rRNA sequencing of fecal samples collected from the two dietary groups at baseline, during oxycodone maintenance (days 1 and 10), 1E, and 2E. There was a significant difference in overall microbial composition between mice on the control diet and mice on the n-3 PUFA-enriched diet (*p*-value < 0.05, R2 = 0.028) after adjusting for study phase (Figure 3A). There was also a significant difference in the microbiome during the oxycodone maintenance phase as compared to both extinction group periods (1E & 2E) while controlling for diet (*p*-value < 0.05, R2 = 0.10). There was no statistical difference between microbiome profiles at day 1 (D1-OXY) of oxycodone maintenance and day 10 (D10-OXY). There was also no statistical difference in overall microbial composition between 1E and 2E. n-3 PUFA supplementation led to a significant increase in species richness measured using the number of types of organisms (Chao1, *p*-value = 0.01), phylogenic diversity measured using the evolutionary distance between organisms (Faith’s PD, *p*-value = 0.003), and species evenness measured using the abundance of organisms across species (Shannon Index, *p*-value = 0.01)(Figure 3B). The mean Bray–Curtis distances between groups were similar overall but it did show a trend, using PreOp as a baseline, where the distance between mice on 10 days of oxycodone and a control diet was larger than the distance between mice on 10 days of oxycodone and an omega-3 enriched diet (*p*-value = 0.1) (Figure 3E).

Differential abundance testing was performed at the genus level to identify microbes that were associated with n-3 PUFA supplementation during the oxycodone maintenance and opioid extinction phases using DESeq2 models at a 5% false discovery rate threshold (*q*-value < 0.05). n-3 PUFA supplementation led to a significant decrease during the oxycodone maintenance phase in *Akkermansia* (4-fold) and *Parabacteroides* (5.5-fold), and a significant increase in multiple genera including *Lactobacillus* (8-fold), *Allobaculum* (7-fold), and *Bifidobacterium* (7-fold) (Figure 4A). During the opioid extinction phase, n-3 PUFA supplementation was similarly associated with the depletion of *Parabacteroides* (3.4-fold) and enrichment of *Bifidobacterium* (8.5-fold), but *Lactobacillus* was not significantly different and additional differences were observed, including increased *Desulfovibrio* (6-fold) (Figure 4B). Similar analysis was then performed to identify genera associated with the opioid extinction phase compared to the oxycodone maintenance phase in one or both diets. Opioid extinction led to a significant decrease in *Akkermansia* (64-fold) and *Bifidobacterium* (8-fold), independent of diet. Opioid extinction also led to a significant decrease in *Parabacteroides* (5-fold), but only within the control diet group (Figure 4C,D). *Lactobacillus* did not significantly change with opioid extinction.

### 3.5. The Activation State of Microglia was Affected by the n-3 PUFA-Enriched Diet but Not Opioid IVSA

To assess the activation state of microglia, we labeled these cells with two antibodies, one against CD68, a lysosomal protein expressed in high levels in activated microglia [58], and one against IBA1, a calcium binding protein found in microglia and macrophages [59]. This enabled us to count the number of CD68-labeled microglia in the dorsomedial and dorsolateral striatum and then assess the shape and density of IBA1 labeling in these CD68-positive cells. A representative slice image is included in Figure 5A.

n-3 PUFA supplementation reduced the number of CD68+ cells as compared to the control diet group (F_(3,16)_ = 14.66, *p* < 0.001; one-way ANOVA), but opioid exposure did not induce any further change (Figure 5B). We next analyzed cell morphology of the CD68+ cells. The cell area of the control diet groups, whether saline or opioid-exposed groups, was lower than that of either n-3 PUFA control (*p* < 0.001) or opioid-exposed (*p* < 0.05) groups (F_(3,18)_ = 6.760, *p* = 0.003, one-way ANOVA). There was no effect of opioid exposure following the control diet, but opioid exposure did lead to a decrease in the cell area in the n-3 PUFA group (*p* < 0.05, n-3 PUFA saline vs. oxycodone) (Figure 5C). Next, we measured the mean grey value or average pixel density of the CD68+ cells. This revealed a decrease in mean density in the n-3 PUFA group compared to the control diet group, regardless of opioid exposure (F_(3,18)_ = 83.17, *p* < 0.001 in each diet group) (Figure 5D). However, there was no effect of opioid self-administration in either diet group (*p* = 0.4 in each diet group) (Figure 5D). We then measured the integrated density of CD68 immunolabeling by taking the sum of the pixels in the standardized selected area within the dorsal striatum. This parameter was similarly altered by diet (F_(3,18)_ = 38.42, *p* < 0.001), with the n-3 PUFA group showing a significant decrease in integrated density (*p* < 0.001), but there was no effect of opioid exposure (*p* = 0.95 and >0.9 in control and n-3 PUFA groups, respectively) (Figure 5E).

## 4. Discussion

We have previously shown that an n-3 PUFA-enriched diet alters the behavioral and cellular adaptations to non-contingent chronic morphine exposure [43]. In this study, we examined how this same dietary regimen altered contingent opioid use. Our novel mouse model behavior paradigm enabled us to assess the effect of volitional opioid exposure across a profile of opioid use that was arguably closer to the human condition. In doing so, we generated a profile of opioid use that differs from most IVSA studies published to date in that we focused on the drug-seeking behaviors during a drug use cycle—extinction, reinstatement, and another period of extinction—to model the phases of abstinence and relapse typical of opioid exposure. Interestingly, we found that oxycodone IVSA established under a short-access FR1 schedule did not result in an increase in the number of active lever presses or reinforcers gained above that of mice receiving saline during the initial maintenance phase, possibly due to an inverted U-dose effect often seen in such studies [60]. However, during an initial extinction period, when the drug was no longer available but all drug-associated cues were present, there was a sharp increase in drug-seeking behaviors irrespective of diet during this first extinction. This drug-seeking declined over subsequent days to a level that was indistinguishable from when the drug was again delivered during the reinstatement phase. The second extinction period showed in the same increase in drug-seeking in the control group, but now the mice on the n-3 PUFA diet failed to show this sharp increase in drug-seeking behavior, which was indistinguishable from that of mice receiving the saline reinforcer.

Our previous study shows how this n-3-PUFA-enriched diet reduces the anxiogenic behavioral profile induced by morphine and striatal glutamatergic signaling [43]. In this study we investigated how this diet could modulate drug-seeking behaviors and the gut microbiome, knowing that there was a microbiome–brain axis that modulates anxiety and depression (review by Foster and McVey Neufeld [61]). We showed that the addition of n-3 PUFAs altered the gut microbiome during opioid exposure and withdrawal, resulting in increased species richness, phylogenetic diversity, and evenness compared to controls. Since dysbiosis (i.e., disease-associated perturbation of the microbiome) is often associated with lower species richness and diversity, n-3 PUFA supplementation may protect against adverse phenotypes mediated by the microbiome such as anxiety-like behavior. This effect is similar to previously published studies examining the role of n-3 PUFA on the microbiome–brain axis. For example, Robertson et al. showed that n-3 PUFA supplementation was associated with enhanced cognition in C57BL/6 mice and an overabundance of *Bifidobacterium* [62]. A second study found that n-3-PUFA-induced enrichment of *Allobaculum* was correlated with a reduction of anxiety-like behavior [43]. Confirming these prior studies, we showed that n-3 PUFA supplementation was associated with an increase in *Bifidobacterium* and *Allobaculum*. While the comparison of microbial composition before and after opioid introduction was inconclusive, we did demonstrate a significant reduction in *Akkermansia* and *Bifidobacterium* with opioid withdrawal. Interestingly, the addition of n-3 PUFA during the extinction phase dampened the reduction of *Bifidobacterium* as compared to the control. Prior research has shown that *Bifidobacterium* may increase the bioavailability of opioids by deconjugating glucuronide in the gut lumen [63]. By having a higher level of deconjugating bacteria, n-3 PUFA may ameliorate the effect of opioid extinction as compared to the control. However, the effect of n-3 PUFA on the IVSA profile during the second extinction phase suggests that the reduction in opioid-seeking was not the result of increased opioid bio-availability alone. One possibility is that n-3-PUFA-induced *Bifidobacterium* and *Lactobacillus* reduced opioid-seeking behavior through microbiome–brain signaling, consistent with literature demonstrating anxiolytic properties of specific strains within these genera [20,64]. n-3 PUFA may also work by suppressing bacteria, such as *Akkermansia*, that can induce anxiety-like behavior [65].

The underlying mechanism causing the increase in drug-seeking behaviors during the extinction cycle is not completely understood and therefore it is also unclear how the n-3-PUFA-enriched diet led to a decrease in these behaviors. Intuitively, it is assumed that the opioid-conditioned response is toward the positive reinforcement associated with drug use; however, negative reinforcement mechanisms can drive the compulsivity of drug addiction and relapse. The negative emotional state that is experienced during drug abstinence following chronic opioid use and reinforcers paired with drug withdrawal have been shown to lead to drug-seeking behaviors [66]. Studies in mice and humans suggest that an elevated level of stress and anxiety increase the probability of relapse and that exposure to stressors reliably reinstates drug-seeking behaviors even after prolonged drug-free periods. This negative emotional state and the associated negative reinforcements may be the driving influence in our study that led the control mice to increase their drug-seeking behaviors during the extinction phases of this behavior paradigm. The reduction in drug-seeking behaviors due to the n-3 PUFA supplementation corresponded to gut microbiome changes during opioid maintenance and withdrawal. The n-3 PUFA supplementation could be playing a protective role in the gut microbiome by mitigating the microbiome changes (i.e., the reduction in *Bifidobacterium*) observed due to opioid withdrawal during the extinction phase. The n-3-PUFA-enriched diet also alleviated the anxiety-associated behaviors in the light–dark test, which was induced by the opioid use and subsequent withdrawal during the extinction period. There is a strong possibility that the effect of n-3 PUFA supplementation seen in the gut microbiome offset the increased levels of anxiety during the forced withdrawal period caused by extinction.

Intermittent noncontingent but not continuous morphine has been shown to alter the gut microbiome and to induce neuro-inflammation. This can be offset by restoring the microbiome to control levels [67]. Considering our findings that the n-3 PUFA diet altered the gut microbiome and offset some of the opioid-induced changes seen during the second extinction phase, we hypothesized that the n-3-PUFA-enriched diet would reduce microglial activation induced by oxycodone self-administration. Assessing the size of the microglia, identified by the presence of CD68 labeling and the shape and intensity of IBA-1 labelled CD68 cells, we found that this diet reduced the basal activation state of microglia, shown as an increase in size and reduction in the number of cells and intensity of the label. However, contrary to our hypothesis, there was little effect of oxycodone self-administration on microglial activation. This may be a result of the low, continuous levels and relatively short access (2 h) of opioid self-administered across the 27 days of the experiment, a regimen that is similar to continuous opioid exposure [67] that may not induce measurable neuro-inflammation. The low sample number of this experiment also warrants caution in furthering our interpretation of this dataset.

This study outlines a profile of opioid self-administration of an initial maintenance phase followed by periods of abstinence, when drug-seeking became pronounced, and may imitate relapse. It was this heightened period of drug-seeking that can be reduced by supplemental n-3 PUFA, which we also show, reduced anxiety. We propose that such periods of opioid exposure and subsequent withdrawal generate a long-term pathological state of anxiety that leads to the high rate of relapse associated with opioid use and that this may be offset by n-3 PUFA supplementation. As n-3 PUFA has beneficial effects on the gut microbiome, this PUFA may offset the effect of opioids on the gut–brain axis. The evidence for this interaction between the gut microbiome and neuropathology continues to grow; we propose that this may be the link by which n-3 PUFA offsets the cellular and behavioral effects of opioids, as shown in this study and our previous work. Additional research does need to be completed in order to understand whether affective-like behaviors and long-term drug-seeking behaviors could be treated by normalizing the intestinal environment or by altering microbial function. Whichever the mechanism involved, our findings do suggest that this supplemental dietary intervention could form part of a treatment protocol for opioid use disorder.

## Figures and Tables

**Figure 1 nutrients-11-01900-f001:**
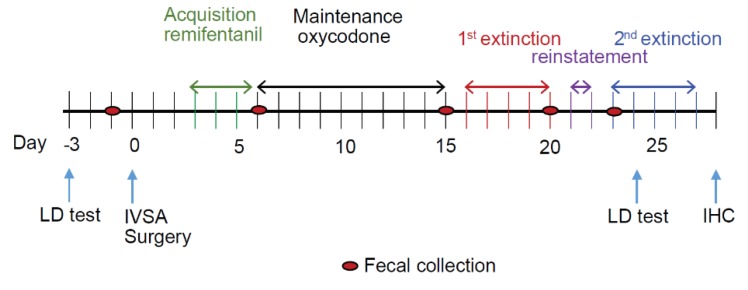
A schematic of the experimental timeline that shows the type and order of interventions in the 27-day protocol used for mice that had received an n-3 PUFA-enriched or control diet for 8 weeks prior to the start of the protocol. Following the implant of the jugular catheter (IVSA surgery), mice underwent daily sessions where, after acquisition of remifentanil self-administration, oxycodone or saline infusion was self-administered for 10 days (the maintenance phase), followed by 5 days of extinction (extinction 1, 1E), in which no further drug was delivered but the same environment and reward-associated cues were presented. This was followed by a 2-day period of reinstatement (reinstatement) in which oxycodone was administered at the same dose and cues as used during maintenance. Thereafter the mice underwent a second extinction (extinction 2, 2E) over the following 5 days under the same conditions as during 1E. Fecal boli were collected before the surgery and at each change in this protocol as indicated. The light–dark (LD) test was conducted before the surgery and on the second day of 2E. Brain tissue was collected at the end of the protocol for microglial analysis using immunohistochemistry (IHC).

**Figure 2 nutrients-11-01900-f002:**
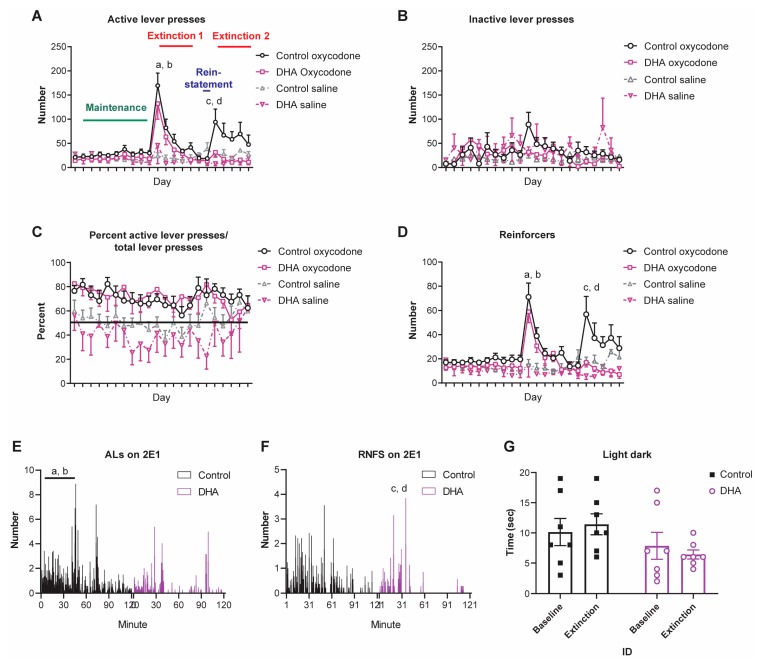
An n-3 PUFA-enriched diet reduces oxycodone seeking behaviors. (**A**) Active lever presses. Mice on both the control diet (black) and the n-3 PUFA-enriched diet (purple) showed an increase in AL pressing on the first day of 1E (a; *p* < 0.0001) and a decline in lever pressing thereafter (b; *p* < 0.0001). AL pressing during the reinstatement did not differ from the last day of 1E training or maintenance. However, during 2E, there was an increase in lever pressing above that of the reinstatement (R1 or R2 vs. 2E1; c; *p* < 0.01) for the control mice but not for the n-3 PUFA mice. Furthermore, active lever pressing by the n-3 PUFA mice was lower than that of the control mice for all days of 2E (d; *p* < 0.05). (**B**) Inactive lever presses. There was no effect of diet or day on this parameter. (**C**) Percent of active lever presses. Although both saline groups showed ≤50% accuracy in AL presses as a percent of total lever presses, this was not different from both opioid groups with ≥50% AL presses. (**D**) Reinforcers. Control diet mice earned more RNFSs on 1E1 (a; *p* < 0.001) and this declined during the remaining days of 1E (b; *p* < 0.0001). The number of RNFSs earned by the controls again increased above that of the reinstatement during 2E (R1 or R2 vs. 2E1; c; *p* < 0.01). n-3 PUFA mice had a similar initial profile as those on the control diet with an increase in RNFS earned on 2E1 (a; *p* < 0.0001), and a decline in RNFS earned for each day of 1E thereafter (b; *p* < 0.0001). However, the 2E1–5 sessions did not induce an increase in RNFSs earned from that seen during the reinstatement. (**E**) Active lever presses during 2E1. During the first hour of this session, the control mice pressed the AL more than the n-3 PUFA mice (a; *p* < 0.05), and this rate declined during the second hour (b; *p* < 0.05). (**F**) Reinforcers during 2E1. During the second but not the first hour, the number of RNFSs earned declined more rapidly in the n-3 PUFA mice than in the control mice (n-3 PUFA vs. control diet; c; *p* < 0.01, and an effect of time in n-3 PUFA but not control mice; d; *p* < 0.01). (**G**) Light–dark test. Although the n-3 PUFA intervention did not alter the basal levels of anxiety, this diet did reduce anxiety, reflected as a quicker entry into the light compartment from the dark compartment by the n-3 PUFA group on 2E2 (a; *p* < 0.05 vs. extinction of the control mice).

**Figure 3 nutrients-11-01900-f003:**
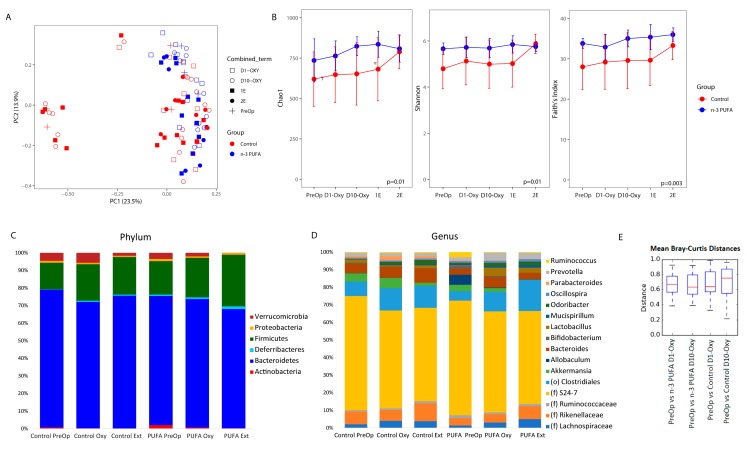
n-3 PUFA supplementation and opioid extinction altered gut microbiome profiles. (**A**) Principal coordinate analysis plot separated by diet and study phase (D1-OXY: Day 1 of oxycodone; D10-OXY: Day 10 of oxycodone; 1E: First extinction phase, 2E: Second extinction phase). Each symbol represents one fecal sample, with color denoting the dietary group and shape denoting the phase. The axis labels indicate the percentage of variation represented by each coordinate. (**B**) Microbial composition and diversity. Species richness (Chao1) (**left**), species evenness (Shannon Index) (**middle**), and phylogenetic diversity (**right**) by diet (control diet in red and n-3-PUFA-enriched diet in blue). (**C**) Phylum and (**D**) genus summary by groups. Bar plots show the average relative abundance of microbes at the phylum and genus levels. D1-OXY and D10-OXY were combined, as were 1E and 2E. (**E**) Bray–Curtis distances between groups.

**Figure 4 nutrients-11-01900-f004:**
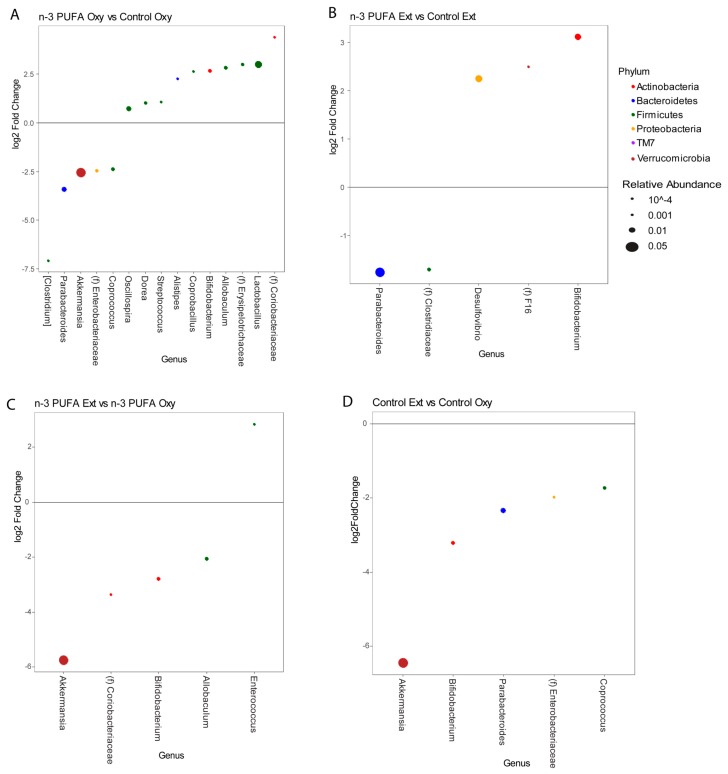
Differential abundance of microbes differed by n-3 PUFA supplementation and opioid extinction. (**A**) Effects of n-3 PUFA on genera abundance during oxycodone maintenance. DESeq2 models were used to identify differentially abundant genera (*q* < 0.05) in the n-3 PUFA supplementation group, combining data from D1-OXY and D10-OXY and adjusting for the time point. Differences are expressed as a log2 fold change, with the dot size proportional to the mean genera abundance across samples and the dot color indicative of phylum. (**B**) Effects of n-3 PUFA on genera abundance during opioid extinction found using DESeq2 analysis of combined 1E and 2E data, adjusted for phase. (**C**) Effects of opioid extinction on genera abundance while on n-3 PUFA supplementation. (**D**) Effects of opioid extinction on genera abundance while on a control diet.

**Figure 5 nutrients-11-01900-f005:**
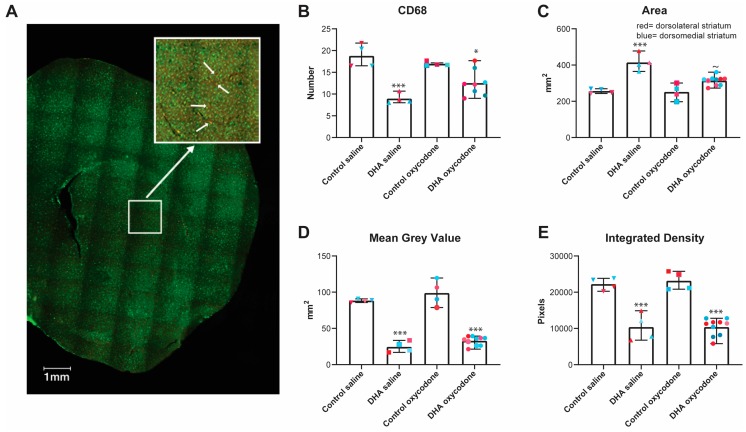
The activation state of microglia was affected by the n-3 PUFA-enriched diet but not by the oxycodone self-administration. To assess the activation state of microglia, the number of CD68-labeled cells in the dorsomedial and dorsolateral striatum were counted and the shape and density of IBA1 labeling in these CD68+ cells was assessed. There was no effect of region on any of these parameters, so the dorsomedial and dorsolateral data were combined. (**A**) A representative example of a coronal section with CD68 labeled in red and IBA1 in green. An example of the dorsolateral region was enlarged, with the arrows showing cells that were labeled with both the CD68 and IBA1 antibodies. (**B**) CD68. The n-3 PUFA dietary regimen reduced the number of CD68+ cells compared to the control group (****p* < 0.001) but opioid IVSA did not induce any further change (**p* < 0.05, n-3 PUFA oxycodone vs. control oxycodone). (**C**) Cell area. The area of the cells of the controls, whether saline or opioid-treated, was lower than that of the n-3 PUFA control (*** *p* < 0.001) or opioid-treated (* *p* < 0.05) groups. The n-3 PUFA opioid-treated group did show a decrease in cell area compared with the n-3 control diet group (*p* ≈ 0.05). (**D**) Mean grey value or average pixel density. This parameter showed an effect of treatment with the n-3 PUFA groups, whether saline or opioid-treated, showing a lower pixel density than the control groups (*** *p* < 0.001 in both groups). (**E**) Integrated density or the sum of the pixels in the selected area. This parameter was similarly altered by treatment with an effect from n-3 PUFA (*** *p* < 0.001), but no effect from opioid exposure.

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
