# Peer review of "Dietary Supplementation with Omega-3 Polyunsaturated Fatty Acids Reduces Opioid-Seeking Behaviors and Alters the Gut Microbiome"

_nutrients, 2019, doi:10.3390/nu11081900_

Round 1

Reviewer 1 Report

The work by Joshua K. Hakimian and colleagues provides a new insight on correlation between dietary supplemented omega-3 polyunsaturated fatty acids and reduction of opioid-seeking behaviors in mice and their relation with gut microbiota. It is also interesting that the dietary supplementation of PUFAs could be used for opioid-dependent anxiety.

 1. It is evident that a long period is required to observe the opioid dependence, withdrawal symptoms and relapses in human.

How could authors justify that the model they designed of only 27 days would be appropriate for the current study?

Reviewer 2 Report

The present study describes the effect of an omega-3 enriched diet on the behavioral and physiological changes, as well as the gut microbiome, in response to a novel self-directed model of chronic opioid exposure. The findings presented in this manuscript demonstrate that an omega-3 diet can ameliorate some of the behavioral and physiological changes caused by chronic opioid exposure. The dietary intervention also changes gut microbial composition, although these differences appear to be parallel to those caused by opioid exposure, rather than counteracting them. The manuscript is well written, and the opioid exposure model is very interesting. The methods are presented in detail and, although complex, quite easy to follow.

However, several sub-figures are missing from the manuscript. Other figures are missing superscripts indicating significance, making results difficult to interpret. There are certain additional methodological errors outlined below, that need to be addressed by the authors. Finally, and most importantly, the dietary intervention is based on a broader omega-3 enrichment, not solely DHA, making the authors claims specific to DHA problematic.

Major Concerns:

The researchers state that ‘the overarching hypothesis for our study is that DHA intake induces a beneficial shift in gut microbiome composition to normalize the genera the become overgrown with opioid self-administration’. The reviewer is not convinced this hypothesis is tested correctly, given that the control and intervention diets differ not only in DHA concentration but also in the presence/absence of EPA. EPA has a known anti-inflammatory component that maybe acting on the gut microbiome independent of DHA. In other words, this study is testing the effect of a broader omega-3 enriched diet, not a diet enriched specifically in DHA.

The reviewer further does not believe this main hypothesis was tested, although this maybe the result of several figures (4C-D) missing from the manuscript. To test this hypothesis, the researchers should compare gut microbiomes of mice treated with Oxy to PreOP microbiomes using, for example, a Bray-Curtis dissimilarity metric. If an omega 3 enriched diet normalizes genera that become overgrown with opioid self-administration, the mean Bray-Curtis distance of mice on the control diet to PreOP should be greater than the mean distance of mice on the omega 3 diet to PreOP (i.e. microbiomes of mice on Oxy fed an omega 3 diet should resemble more closely the microbiomes of mice before Oxy administration than microbiomes of mice on Oxy fed a control diet.)

Do the researchers have bio banked blood samples from the mice used in these studies? These samples could be used to test whether opioid blood levels vary across diets (potentially due to changes in deconjugation capacity of the gut microbiota, as suggested by the authors in the discussion).

Along the same lines, measuring DHA and EPA blood levels in the blood would confirm a successful dietary intervention. Alternatively, the authors should state in the text that their dietary intervention resulted in changes in brain DHA levels in previous studies from the group, confirming the validity of the intervention.

In the methods section, the researchers report that a few (how many?) mice were affected by an overgrowth of segmented filamentous bacteria. This genus was excluded from differential abundance analysis. Although it is understandable to exclude this genus from analysis using DESeq2, such contamination will affect both alpha and beta-diversity analyses performed separately in this study. The researchers should elaborate on how contaminated samples were handled in alpha and beta-diversity analyses, not just DESeq2.

For gut microbiome 16S RNA sequencing, can the researchers justify the different number of animals used at each time point? Why is there considerable variation?

Analysis in Figure 3A should be replicated using a more commonly applied dissimilarity metric in the field (Bray-Curtis or Weighted Unifrac). Alternatively, a justification for using square root Jensen-Shannon divergence over other metrics should be provided.

In Figure 3B, there is an increased alpha-diversity in all metrics tested across the two extinction phases relative to the PreOp time point. In particular, in the 2E phase, alpha-diversity appears significantly higher than PreOp in the control diet mice, and no longer significantly different from the omega 3 diet mice. What makes this result even more interesting is that the 2E phase is where some of the beneficial effects of omega-3s on behavioral outcomes are observed, even though gut microbial diversity between diets becomes more similar at this time point. The authors do not discuss or analyze this result in their manuscript. Additional insight into this very interesting observation would improve the paper.

Figures 4C-D are missing from the manuscript.

In lines 422/423 the researchers write “the cell area of the control diet group, whether saline or opioid-treated was lower than that of the DHA control (p<0.001) or opioid self-administration (p<0.05) groups.”  This sentence is confusing and Figure 5C it’s referring to has no symbols representing significance between groups, making interpretation even more difficult.

Minor Concerns:

Line 50: the word “and” is missing after the word “fentanyl”.

A whole phrase within a sentence beginning in line 467 and ending in 469 is repeated twice and should be corrected.

In lines 122-123 the researchers write “Where possible, experimenters were blind to treatment and diet for both behavioral testing and data analyses.” It would be beneficial if the researchers be more specific about the blinding procedure conducted in their study.

Figure 2G is missing the letter “a” demonstrating significance, referenced in the figure description.

In lines 352 and 354 the researchers report a significant difference in overall microbial composition between two diet groups (line 352) as well as the oxycodone maintenance and extinction phase (line 354). Given this analysis was performed using PERMANOVA, the effect size (R2) should be reported.

In Figure 5, the ‘***’ symbols referred to in the figure description to highlight significance are not shown in the figures themselves. This should be corrected.

Round 2

Reviewer 2 Report

The authors have addressed all my concerns.